# Varroa Mite Counting Based on Hyperspectral Imaging

**DOI:** 10.3390/s24144437

**Published:** 2024-07-09

**Authors:** Amira Ghezal, Christian Jair Luis Peña, Andreas König

**Affiliations:** Fachbereich Elektrotechnik und Informationstechnik, Lehrstuhl Kognitive Integrierte Sensorsysteme, RPTU Kaiserslautern, 67663 Kaiserslautern, Germany; ghezal@eit.uni-kl.de (A.G.); pena@rhrk.uni-kl.de (C.J.L.P.)

**Keywords:** automated Varroa counting, hyperspectral imaging, shape detection, segmentation

## Abstract

Varroa mite infestation poses a severe threat to honeybee colonies globally. This study investigates the feasibility of utilizing the HS-Cam and machine learning techniques for Varroa mite counting. The methodology involves image acquisition, dimensionality reduction through Principal Component Analysis (PCA), and machine learning-based segmentation and classification algorithms. Specifically, a k-Nearest Neighbors (kNNs) model distinguishes Varroa mites from other objects in the images, while a Support Vector Machine (SVM) classifier enhances shape detection. The final phase integrates a dedicated counting algorithm, leveraging outputs from the SVM classifier to quantify Varroa mite populations in hyperspectral images. The preliminary results demonstrate segmentation accuracy exceeding 99% and an average precision of 0.9983 and recall of 0.9947 across all the classes. The results obtained from our machine learning-based approach for Varroa mite counting were compared against ground-truth labels obtained through manual counting, demonstrating a high degree of agreement between the automated counting and manual ground truth. Despite working with a limited dataset, the HS-Cam showcases its potential for Varroa counting, delivering superior performance compared to traditional RGB images. Future research directions include validating the proposed hyperspectral imaging methodology with a more extensive and diverse dataset. Additionally, the effectiveness of using a near-infrared (NIR) excitation source for Varroa detection will be explored, along with assessing smartphone integration feasibility.

## 1. Introduction

Within the intricate sphere of agriculture, honeybees serve as vital contributors, orchestrating the necessary pollination for the health and productivity of diverse crops [1,2]. However, this delicate balance faces a powerful adversary in the form of the Varroa mite (Varroa destructor), a small but impactful parasite that belongs to the family of mites and presents a significant challenge to beekeepers worldwide [3]. The consequences of the Varroa mite extend beyond the limits of the apiary, disturbing the complex network of ecological connections reliant on pollination. Conventional counting approaches frequently lack the accuracy needed for targeted intervention and can also impose a monotonous routine for beekeepers [3,4]. To fill this gap, our study aims to present an innovative methodology in Varroa mite counting, achieved through the combination of hyperspectral imaging with machine learning classification. Moreover, the integration of hyperspectral imaging in agriculture has been extensively investigated, emphasizing the technology’s diverse potential in the field. Hyperspectral imaging, coupled with advanced machine and deep learning approaches, has demonstrated success [5,6]. Its effectiveness in assessing the quality of agro-food products is well established [7]. Furthermore, other research explores hyperspectral imaging applications, such as determining fruit ripeness [8], employing near-infrared spectroscopy for inspecting grape quality [9], detecting cancer [10], investigating works of art [11], assessing the quality of fruits and vegetables [12,13], applications in medicine [14], and evaluating the ripeness of Achacha Fruit [8]. The necessity for accurate Varroa mite counting goes beyond technological obstacles, representing a crucial aspect of sustaining healthy honeybee colonies. The current counting techniques, frequently dependent on manual and labor-intensive procedures [4], emphasize the urgent requirement for inventive, automated solutions.

Diverging from previous research, Divasón et al. [15] focus on employing deep learning techniques with smartphone-captured images for Varroa detection, while Noriega et al. [16] propose a method using Legendre–Fourier moments for Varroa classification, emphasizing distinctive descriptors for RGB image representation. Similarly, Sevin et al. [17] introduce Var-Gor, a physical device for Varroa detection attached to hive entrances, all of which utilize color images from traditional cameras and detect Varroa from bees rather than from the Varroa board. In contrast, our approach deviates by integrating hyperspectral imaging with machine learning classification, enabling us to capture detailed spectral information across multiple bands. This enhances the precision and accuracy of Varroa identification and counting, empowering beekeepers with a timely and informed decision-making tool. In comparison to deep learning, which typically requires large datasets and excels at automatically learning complex patterns, machine learning can operate with smaller datasets and offers more interpretability. Furthermore, previous studies utilizing visual range and RGB sensing have established that the red channel exhibits the highest sensitivity for Varroa mite detection [15,18]. The HS-Cam used in this paper benefits from its focus on the visual red and near-infrared range. The camera series is MV1-D2048x1088-HS02-96-G2, manufactured by Photonfocus AG. based on the IMEC CMV2K-SM5x5-NIR CMOS image sensor, located in Bahnhofplatz 10. CH-8853 Lachen SZ. Switzerland. Its multi-channel resolving capability helps to better avoid misdetection of non-Varroa items within the wide visual red range. Moreover, while systems like IndusBee 4.0 [19] and the BeE-nose system [20] focus on various sensor integrations for hive monitoring and gas-sensing capabilities within the hive environment, our hyperspectral imaging approach concentrates on direct Varroa counting from the Varroa board. Furthermore, while the in-hive soft sensor approach [21] aims to estimate Varroa infestation levels based on indirect cues from hive activity patterns, our research emphasizes the development of a targeted solution for Varroa counting, providing beekeepers with an automated and comprehensive solution that surpasses the limitations of existing techniques: the manual methods. By directly counting Varroa destructors from the Varroa board and providing actionable data, our approach enables more effective pest management strategies.

In the subsequent sections of this paper, we will present the materials and methods utilized in this study, delineate the methodology employed for hyperspectral imaging-based Varroa mite counting, report the results of our investigation, and discuss the implications and potential applications of our findings. Finally, we will conclude with a summary of our key contributions and propose avenues for future research in this domain.

## 2. Materials and Methods

### 2.1. Image Acquisition Setup and Conditions

In our study, we utilize a hyperspectral imaging (HSI) sensor to capture detailed spectral information across a range of wavelengths. The [camera model MV1-D2048x1088-HS02-96-G2-10] hyperspectral sensor, manufactured by Photonfocus [22], incorporates the IMEC snapshot mosaic CMV2K-SSM5x5-NIR sensor, featuring a 5 × 5 mosaic arrangement of filters optimized for hyperspectral imaging. In this configuration, each pixel contains a mosaic of 25 channels in the NIR range, allowing for the simultaneous capture of spectral information across multiple narrow wavelength bands. This sensor operates on the principle of capturing reflected or emitted light from objects in the scene, allowing for the extraction of spectral signatures unique to different materials or substances. The hyperspectral camera is equipped with a set of band-pass optical filters from Edmund Optics (975 nm Shortpass and 675 nm Longpass (O.D. 4)), and the sensor boasts specifications that include a spatial resolution of 2048 × 1088 pixels and a wavelength range of 675 to 975 nm. These specifications, coupled with high sensitivity and dynamic range, make the sensor well suited and enable a detailed analysis of the spectral signatures associated with Varroa mites and their surroundings, facilitating accurate identification and counting.

Furthermore, to provide a comprehensive overview of the sensor’s performance characteristics, we have summarized its specifications in Table 1.

The two Varroa boards utilized in this study were obtained from the bee lab hives of the IndusBee 4.0 activity in August 2022 and August 2023. The Varroa board, also referred to as a mite drop board, is a simple essential object used by beekeepers to assess Varroa mite infestations in honeybee colonies. As shown in Figure 1, the Varroa board is inserted like a drawer in the bottom unit of a beehive, where it serves to collect fallen mites. These mites, dislodged from adult bees or developing brood, land on the surface of the board as they fall, providing a visual indication of the level of Varroa mite infestation within the colony. The Varroa board plays a crucial role in Varroa mite monitoring programs, enabling beekeepers to gauge the severity of infestations and make informed management decisions to protect honeybee health. Throughout our research, we utilize Varroa boards as part of our data collection methodology to quantify Varroa mite populations and evaluate the effectiveness of our hyperspectral imaging-based counting method.

Illumination is provided by two 50 W halogen lamps positioned 28 cm above the board at an angle of approximately 45 degrees to the camera. Despite the spectral dataset provided by the sensor, each image’s coverage is inherently limited to a region size of 1.85 cm × 3.55 cm, representing only 0.26% of the entire board as depicted in Figure 2. A total of 2,228,224 pixels per region contribute to the detailed hyperspectral image capture process. To ensure accuracy and quality control in our analysis, a manual verification step is implemented. To address the limitations of the current setup and improve image acquisition, our goal is to enable broader region coverage and more efficient mite identification. Drawing inspiration from numerous studies utilizing a roller mechanism for scanning the entire board [23], we aim to incorporate a scanning approach into our methodology and capture panoramic images as a solution to cover the entire sample efficiently. Furthermore, building upon insights from a related study [24], we plan to enhance Varroa mite identification efficiency by incorporating an alternative light source, such as near-infrared (NIR) illumination. This strategic approach aims to further improve the automated counting process by better highlighting Varroa mites and continually refining our detection capabilities.

### 2.2. Recognition Pipeline and Calibration Process

This section serves as an introduction to the designed recognition pipeline tailored for analyzing multispectral data captured by the HS-Cam, presenting an overview of its structure and fundamental components. Building upon the commonly used approach for multispectral image analysis, our pipeline is extended to accommodate the specific characteristics of data acquired from the HS-Cam. Figure 3 visually represents the recognition pipeline, providing a snapshot of its architecture and highlighting the extensions made to incorporate HS-Cam-specific data processing techniques. This includes integrating implemented shape detection techniques, such as a Support Vector Machine (SVM) classifier, to improve the identification accuracy of Varroa mites based on their distinctive shapes.

Our recognition pipeline involves a series of pivotal steps. The process commences with calibration, a trilogy comprising reflectance calculation, demosaicing, and spectral correction (Figure 4). These calibration steps play a crucial role in guaranteeing the accuracy and standardization of hyperspectral data.

Camera calibration is conducted following the manufacturer’s specifications provided by Photonfocus (Model MV1-D2048x1088-HS02-96-G2-10). We employ Python programming language along with OpenCV and NumPy libraries for image processing tasks and XML parsing. Here is a concise summary of each step: Reflectance calculation is a fundamental step in the calibration process. It normalizes the spectral intensity, accounting for variations in illumination conditions. For this, we calculate the reflectance using a ratio-based method, leveraging images of a white reference and a dark background. The reflectance values are computed based on the ratio of the target and reference images, adjusted by a calibration factor. In Equation (Equation 1), “Target” represents the measured spectral intensity of the target object, “Dark” is the spectral intensity of the dark reference, and “White” is the spectral intensity of a white reference. This normalization ensures accurate and standardized hyperspectral data.

Equation (Equation 1) describes the calculation of reflectance using the following formula: (1)Reflectance(R)=(Target−Dark)/(White−Dark)

The calculated reflectance provides a normalized measure, accounting for variations in illumination conditions.

In the demosaicing process, the reflectance image undergoes transformation into a three-dimensional cube. This cube is constructed by assigning each pixel’s reflectance value to a specific channel based on its position. Subsequently, a demosaicing operation is performed on each channel, involving the linear interpolation of missing values along rows and columns, resulting in a demosaiced cube that provides a more detailed and complete representation of the reflectance data for further analysis.

Spectral correction enhances the original pixel spectrum. This process entails the utilization of a correction matrix, which is an essential component of the standard calibration method provided by camera manufacturer Photonfocus and sensor chip manufacturer (IMEC); it aims to mitigate the effect of the overlapping between the channels. The calibration matrix is constructed by extracting coefficients for each virtual band provided in the XML file obtained from the IMEC sensor chip manufacturer, which are then organized into a matrix format [25]. Each row of the matrix represents coefficients for a virtual band. This matrix acts as a transformation matrix, facilitating the adjustment of sensor readings for spectral correction. Once obtained, the correction matrix is applied to sensor readings through matrix multiplication as outlined in Equation (Equation 2). This process ensures that each pixel’s spectrum is appropriately adjusted based on the coefficients specified in the correction matrix. Equation (Equation 2) introduces a correction process for the obtained spectrum:(2)$$\mathrm{Corrected}\;\mathrm{Spectrum}=\mathrm{Original}\;\mathrm{Pixel}\;\mathrm{Spectrum}\times \mathrm{Correction}\;{\mathrm{Matrix}}^{T}$$

This equation represents a correction step to refine the original pixel spectrum.

Once the images are calibrated, our pipeline advances to a crucial stage: pixel-wise supervised segmentation. This step is pivotal as it isolates Varroa mites from other elements in the image, forming the foundation for the subsequent shape detection phase. Varroa mites, characterized by a distinct shape, are counted based on the shapes identified in the segmented images. This two-phase approach ensures a systematic and efficient methodology for Varroa counting.

The segmentation process entails a series of steps. Initially, the hyperspectral image undergoes preprocessing, including dimensionality reduction via a Principal Component Analysis (PCA) from the scikit-learn library in Python. This step reduces the computational complexity while preserving essential spectral information. Python, alongside OpenCV and NumPy, facilitate efficient data manipulation throughout the process. For classification, we implement a k-Nearest Neighbors (kNNs) classifier, which is trained to classify each pixel belonging either to a Varroa mite, or to non-Varroa pattern based on spectral characteristics, using the scikit-learn library. Following segmentation, additional image enhancement techniques are applied, such as bilateral filtering from OpenCV, to diminish noise and refine the segmentation quality.

In the shape counting process, the geometric and morphological features are extracted from the images to characterize the shapes using functions provided by the OpenCV library in Python. The contours of the shapes in the image are identified using the cv2.findContours() function, which detects the outlines of objects. Then, various geometric and morphological features are computed and saved for each contour to characterize the shapes. Feature selection follows using a Random Forest classifier in conjunction with the SelectFromModel method from the scikit-learn library in Python. This step involves training the classifier on the dataset to determine the importance of each feature. Subsequently, a feature selector is created based on the median feature importance, retaining only the most significant features for further analysis. For classification, a Support Vector Machine classifier (SVM) from the scikit-learn library is employed to predict whether each identified shape represents a Varroa mite or not. Once trained, the classification model is applied to the segmented shapes to determine if they correspond to Varroa mites based on their geometric and morphological features. Finally, the predicted labels are overlaid on the original images alongside the contours of the identified shapes, providing a visual representation of the classification results. Throughout the process, the implementation of these techniques benefits from the use of OpenCV and NumPy libraries.

### 2.3. Labeling and Data Generation

In the feature extraction and label generation phase of our methodology for pixel-wise supervised segmentation, hyperspectral images are manually labeled to distinguish between Varroa and non-Varroa classes. This process involves identifying Varroa mites and delineating them from the background and debris regions within the images. We define the mask regions based on the ground truth provided by human experts. The Varroa mites are labeled by coloring the corresponding regions white, while the remaining areas are delineated and thresholded to black to represent the background and debris. The same process is applied to label the non-Varroa class. This careful labeling procedure ensures the creation of masks accurately representing the presence and location of both classes, forming essential binary maps for subsequent segmentation and classification tasks. We collected a total of 12 images from the Varroa board dated August 2023, using the HS-Camera, representing approximately 4.95% of the entire board. The dataset generated contains 6 images, which correspond approximately to 2.47% of the total data collected. Figure 5 provides a visual representation of the labeling process, illustrating the originals and the masks created for both classes. The labeling process involves manual annotation by creating masks to delineate Varroa and non-Varroa regions. These masks are then utilized for labeling the data. For testing, an unseen image is processed, and the resulting segmented output indicates Varroa presence in areas where Varroa was not present in the original image. These images are captured from different regions of the Varroa board. In an effort to ensure uniformity, we aim to maintain an equal number of pixels for both Varroa and non-Varroa masks from each image. However, during model testing, an imbalance in the pixel values between the Varroa and non-Varroa masks can be observed, resulting in a segmented image where many background pixels are erroneously considered as Varroa by the model. To address this issue and increase segmentation accuracy, additional pixels are added to the non-Varroa mask, as shown in the first image depicted in Figure 5.

Figure 6 provides a visual summary of our training dataset analysis, utilizing Principal Component Analysis (PCA) to represent the training dataset derived from six images displayed in Figure 5.

This representation is obtained after implementing a dual dataset split to accelerate the processing. We first extract 80% of the data for analysis and then partition it into training and testing sets, with 90% allocated for testing. In our analysis, we utilize 5 principal components, which collectively explain approximately 93.58% of the total variance in the dataset. PCA enables us to condense the original hyperspectral data, consisting of 25 channels, into a smaller set of components while retaining the most important information. In the Varroa shape analysis phase, segmented images are processed to extract essential features. Varroa shapes are labeled as Class 1, while shapes with different parameter values are labeled as Class 2, resulting in a total of 11 extracted features. For feature selection, we utilize the SelectFromModel method from the sklearn library. This method employs the Random Forest classifier, trained with 100 decision trees, to identify the most informative features based on their importance scores. Six features, including the Area, Perimeter, Radius, Convex-hull-area, Solidity, and Circularity, are selected to be the most valuable for characterizing Varroa shapes. Figure 7 illustrates the PCA visualization of the dataset used in the shape detection phase. The left subplot shows the distribution of instances in a 2D space obtained through a Principal Component Analysis (PCA) for the training set, while the right subplot illustrates the distribution for the test set. Instances belonging to the “Varroa” and “Non-Varroa” classes display distinct clusters, indicating the potential separability between the classes in the reduced space. However, it is important to note that this analysis is based on a small dataset.

## 3. Results

Within this section, we present a framework to evaluate the effectiveness of our recognition pipeline in Varroa counting using hyperspectral imaging. For the segmentation phase, we utilized a dataset comprising six images (depicted in Figure 5) to extract the features for two classes: “Varroa” and “Non-Varroa”. The same set of six images was used for extracting the pixel values of both classes.

This dataset was divided twice using the train–test split function to extract 80% of the dataset and then the second splitting was used on this 80% to allocate 10% for training and 90% for testing. The training set consisted of 14,210 samples, each with the same shape (5, 2), while the test set comprised 127,894 samples, also with the same shape (5, 2). The six remaining images, not included in the generated dataset, were employed to test the model’s ability to classify the two classes and generate segmented images. This testing phase was distinct from the overall assessment of the classification accuracy on the 80% test set. Both methods served as evaluation techniques for assessing the model’s performance. Then, we selected the k-Nearest Neighbors classifier (kNNs) with n=1, leveraging its efficacy for pixel-wise classification during testing, when compared with the other classifier as reported in Table 2.

In Table 2, the reported results were obtained following a dual dataset split aimed at expediting the processing. Each image designated for segmentation underwent resizing from its original dimensions (1088, 2048) to updated ones (200, 400) using OpenCV’s resize function before being provided to the kNN classifier. This resizing process involved adjusting the dimensions while preserving the content through interpolation. Each channel of the original image underwent interpolation for downsampling. These measures were implemented to enhance the computational efficiency and reduce the processing time. In Figure 8, we present the segmentation outcomes of the 12 images. After generating the initial segmentation result from the classifier, the image underwent further enhancement using a bilateral filter. With a kernel size of 9 and spatial and color sigmas set to 75, the filter aimed to enhance the shape of the Varroa while preserving important details for the subsequent shape analysis. With a kernel size of 9 and a spatial and color sigma of 75, the filter effectively smoothed blobs in the image while retaining the structural integrity of the Varroa.

Following the enhancement step, a binary mask representing the detected regions of interest, particularly Varroa areas, was created based on the contours, which highlights Varroa regions with white pixels against a black background. Additionally, we computed confusion matrices to gain a comprehensive understanding of the class distribution. Below are the confusion matrices for the training and test sets:
**Training Set Confusion Matrix:**


$$\mathrm{Confusion}\;\mathrm{Matrix}\;(\mathrm{Trainset})=\left[\begin{array}{cc} 1244 & 0\\ 0 & 12,966 \end{array}\right]$$



**Test Set Confusion Matrix:**



$$\mathrm{Confusion}\;\mathrm{Matrix}\;(\mathrm{Testset})=\left[\begin{array}{cc} 10,695 & 89\\ 23 & 117,087 \end{array}\right]$$


Moving to the shape detection phase, the dataset consists of 11 images, with a training set size (92, 6, 2) and a test set size (24, 6, 2). A Support Vector Machine (SVM) with a kernel function and C=1 was employed. Given that the dataset encompasses data from 11 images, using the train–test split function from the sklearn library, 80% of it was employed for training, leaving the remaining 20% for model evaluation as a test set. In addition, one test image (Img4 in Figure 8) was kept for shape analysis classification, serving as a targeted evaluation of the model’s Varroa counting ability, separated from the overall classification accuracy assessment on the remaining test set. Both evaluation approaches contribute to determining the model’s performance. In Table 3, we present the test set accuracy and cross-validation accuracy for each classifier, alongside the selected hyperparameters obtained through a meticulous grid search. The classifiers underwent training and evaluation using both cross-validation and test set accuracy metrics. Cross-validation served to assess the model robustness and mitigate overfitting risks by evaluating performance across multiple data subsets.

Figure 9 displays the output of the 12 images, showcasing the detection of Varroa mites in each image alongside the ground-truth count.

To preliminarily evaluate the effectiveness of the HS-Cam’s capability and its impact on the overall success of our Varroa counting methodology, we performed a comparative analysis of the hyperspectral imaging versus traditional 3-band imaging using various methods, including Principal Component Analysis (PCA), HSI2RGB conversion, and Gaussian functions. These methods were employed to generate RGB-like images, which were then used in the segmentation phase against the hyperspectral image outputs. Figure 10 visually illustrates the comparisons between the segmentation results of the hyperspectral images and the generated RGB-like images.

The quantitative metrics are detailed in Table 4, presenting a comparative evaluation of the segmentation methods based on the hyperspectral imaging (HSI) and RGB-like images.

## 4. Discussion

This section offers a comprehensive analysis and interpretation of our findings. The decision to use the kNN and “Linear” SVM for pixel-wise segmentation and shape analysis was made after a thorough evaluation of their theoretical suitability. The “Linear” SVM, known for its proficiency with small datasets, was beneficial for the shape analysis due to its ability to find optimal hyperplanes and prevent overfitting. Conversely, kNN’s simplicity and effectiveness in classifying data based on the similarity to neighbors suited segmentation tasks well, and it is robust to noise and outliers. Both models consistently demonstrated superior performance in segmentation accuracy and shape classification, as evaluated by metrics such as the accuracy, precision, recall, and F1-score.

An analysis of Table 2 revealed that while the kNN classifier excelled in both the training and testing phases, the segmentation analysis in Figure 8 unveiled notable background misclassifications, attributed to the overwhelming dominance of Class 2, comprising 91.25% (12,966 samples) of the dataset compared to Class 1’s 8.75% (1244 samples). This highlights the dataset’s imbalance, necessitating computation of confusion matrices for comprehensive insight. All instances of Class 1 and Class 2 are correctly classified, evidenced by the true positives for Class 2 (12,966) and true negatives for Class 1 (1244). The absence of false positives and false negatives (both are 0) suggests that the model is perfectly accurate on the training data, indicating potential overfitting to the training set as real-world data often contains some noise and variability.

Similarly, the test set confusion matrix reveals a high accuracy with a slight bias towards Class 2. The number of true positives (117,087) is significantly high compared to true negatives (10,695). The low number of false positives (89) for Class 2 indicates strong prediction accuracy, while the misclassification of Class 1 instances (23 false negatives) suggests a minor imbalance. This slight bias toward Class 2 is evident, though the classifier maintains a high overall accuracy across both classes. The precision, recall, and F1-score metrics have been also observed for evaluation. The kNN classifier with *n* = 1 demonstrated a precision of 0.9978, a recall of 0.9917, and an F1-score of 0.9947, affirming its reliability amidst misclassifications. Conversely, the SVM classifier with an “rbf” kernel exhibited perfect identification of negative instances but failed to detect any positive instances, resulting in an F1-score of 0.0. In summary, while some images showed misclassified the pixels of the background and debris (Figure 8), all exhibited well-defined Varroa shapes that were correctly identified. This precise characterization of blob shapes is crucial for subsequent shape analysis phases.

For the shape analysis, the results in Table 3 demonstrated high accuracies for all the classifiers on both the cross-validation and test sets. However, the small dataset size may have influenced these results, potentially leading to overfitting. Among these classifiers, SVM achieved the highest accuracies on both evaluation metrics, indicating its suitability for shape analysis tasks due to its ability to avoid overfitting and generalize effectively. Figure 9 validated the effectiveness of the SVM classifier in accurately quantifying Varroa mites in the images, reinforcing the advantages of our approach in Varroa mite counting using hyperspectral imaging.

The use of hyperspectral imaging with the HS-Cam proved to be transformative for Varroa segmentation. Particularly with Img4 in Figure 8, applying a threshold to the segmented output revealed precise Varroa shapes outlined against a black background (Figure 10). In contrast, the RGB-like images exhibited misclassification issues, particularly with debris being inaccurately identified as Varroa, canceling the benefits of applying a threshold (Area > 140), leading to noise in the segmented images. These limitations underscored the superiority of the HSI method over HSI2RGB, as highlighted by the quantitative metrics in Table 4, including the precision, recall, F1-score, and accuracy percentages. This evaluation emphasizes the effort of integrating the HS-Cam into our methodology, resulting in a substantial improvement in Varroa counting accuracy. However, alongside these benefits, it is imperative to acknowledge the specific challenges and limitations associated with implementing hyperspectral imaging for Varroa counting scenarios.

One such challenge arises when the camera fails to capture the complete shape of a Varroa mite, complicating shape detection due to incomplete representation in segmented images. Additionally, Varroa hidden below debris poses another obstacle, impeding accurate segmentation. Addressing these challenges is essential for gaining a comprehensive understanding of the capabilities and constraints of hyperspectral imaging in Varroa counting scenarios.

The utilization of the HS-Cam in Varroa mite counting justifies the associated effort and cost, demonstrating the advantages of automating the process. This automation significantly reduces the time and effort required for counting. The HS-Cam improves the accuracy and efficiency compared to the traditional RGB imaging methods due to its spectral information while also ensuring consistent and untiring recognition performance across various apiaries and providing uniform expert-level results regardless of the hive owner or operator. While the HS-Cam serves as the model for the discussed automation, similar benefits may be achievable with other cameras used in the research. However, it is important to recognize limitations such as the lack of hand–eye coordination, which may impact decision-making in ambiguous situations.

## 5. Conclusions

In conclusion, our study demonstrates the feasibility of our proposed method, which combines hyperspectral imaging technology with machine learning methods for accurate Varroa mite counting. The acquisition of detailed spectral information enables Varroa segmentation with well-defined shapes with minimal misclassifications compared to traditional RGB-like images. Our proposed method highlights the benefits gained through automation, reducing reliance on labor-intensive counting, and simplifies Varroa counting processes in an efficient and accurate way beyond traditional practices. Looking ahead, our approach’s adaptability to confined spaces addresses limitations posed by a restricted region size. We propose exploring the collection of panoramic data, akin to capturing the whole board, to enhance the comprehensiveness of Varroa detection. Additionally, we envision a potential future where HS-Cam technology seamlessly integrates into smartphones, akin to the transformative integration of infrared (IR) technology into Apple devices. In summary, the first investigation of our approach with HS-Cam/data gives encouraging results. However, it is worth noting that deep learning (DL) was considered but not pursued, as the required number of training examples is hard to obtain in this application. If these results can be repeated or even improved for larger-scale data and more sophisticated acquisition scenarios and algorithmic processing, a valuable advance in Varroa detection and monitoring can be expected.

## Figures and Tables

**Figure 1 sensors-24-04437-f001:**
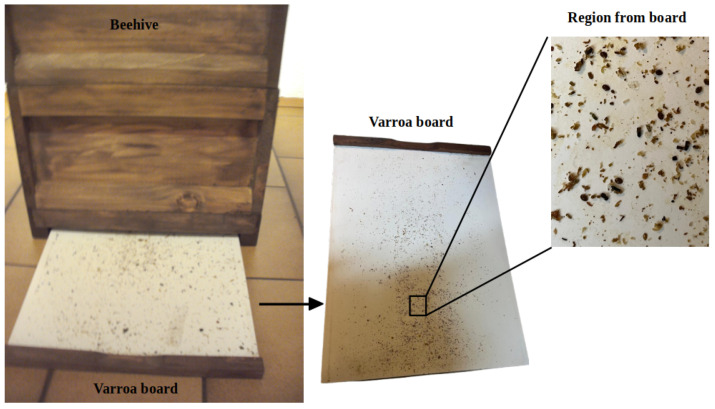
Beehive with half-drawn Varroa board for Varroa mite monitoring. Inset: Close-up of Varroa mites with debris captured on the board.

**Figure 2 sensors-24-04437-f002:**
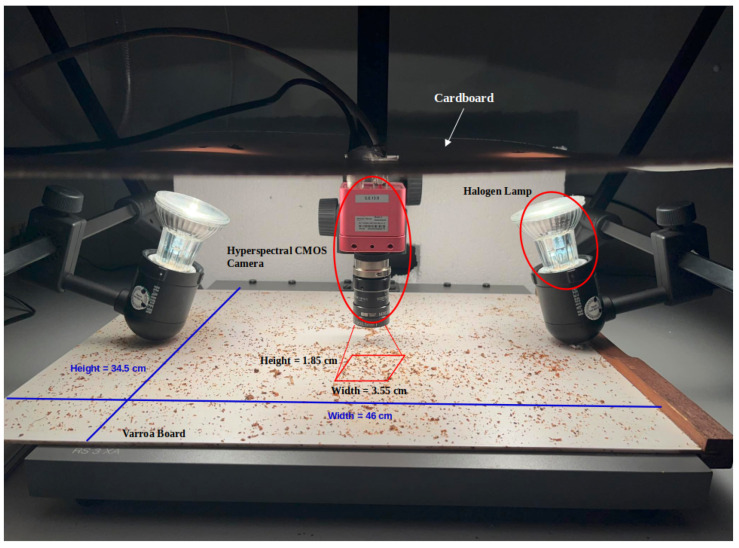
Hyperspectral acquisition setup with Varroa board and selected ROI.

**Figure 3 sensors-24-04437-f003:**
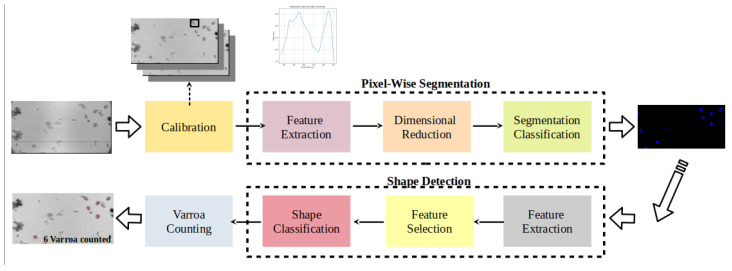
Recognition pipeline.

**Figure 4 sensors-24-04437-f004:**
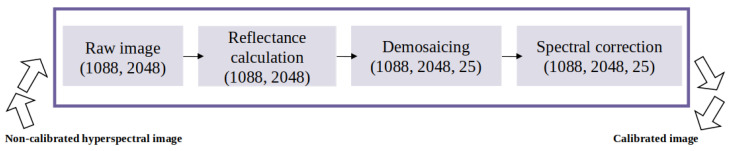
Calibration process.

**Figure 5 sensors-24-04437-f005:**
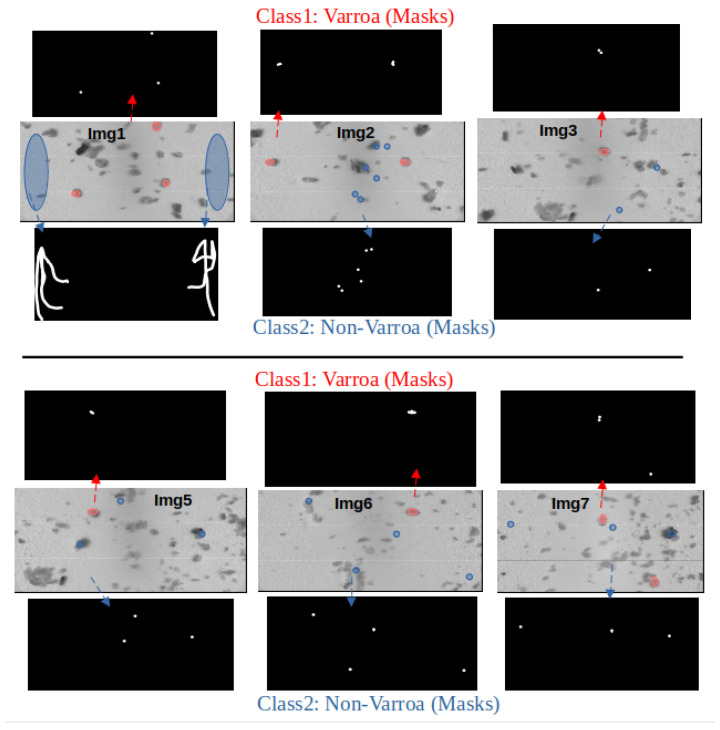
Illustration of masks and images used for labeling of two classes. Masks highlight regions of interest in images for precise labeling of Varroa mites and non-Varroa, aiding in data annotation. The arrows in the figure indicate specific regions where the masks are applied, guiding the viewer to the areas of interest: blue arrows lead to non varroa region, and red arrows lead to varroa region.

**Figure 6 sensors-24-04437-f006:**
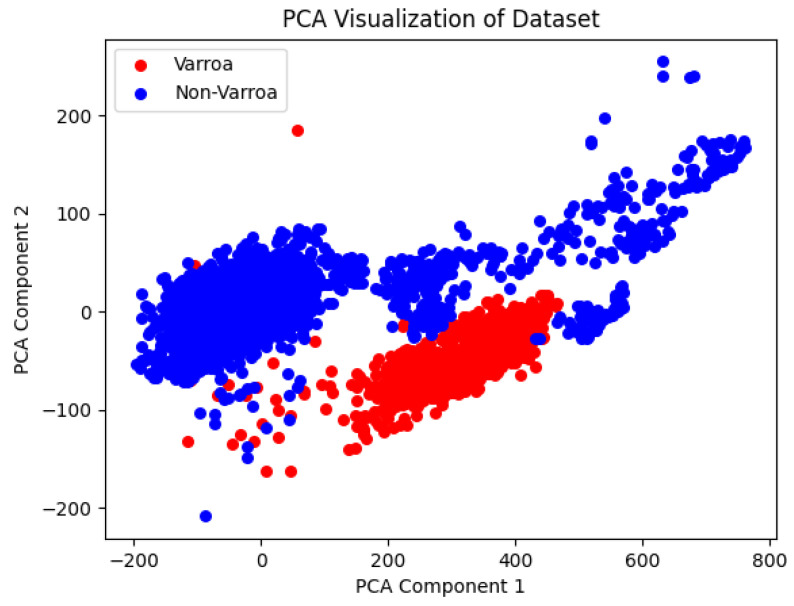
PCA visualization illustrates sample distribution from two classes, revealing dataset separability.

**Figure 7 sensors-24-04437-f007:**
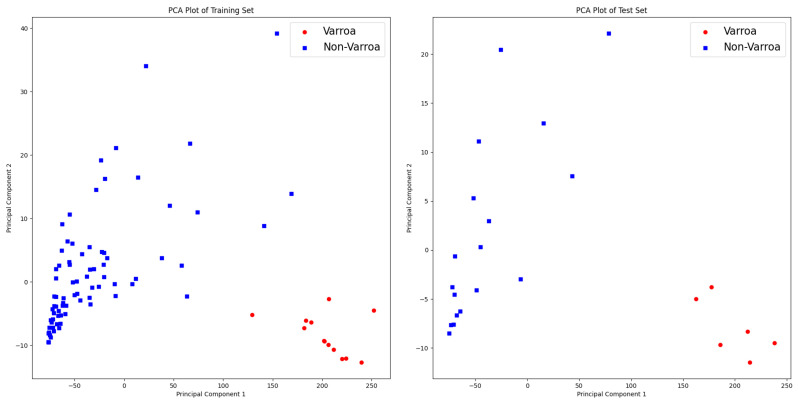
PCA visualization of training and test datasets for shape analysis.

**Figure 8 sensors-24-04437-f008:**
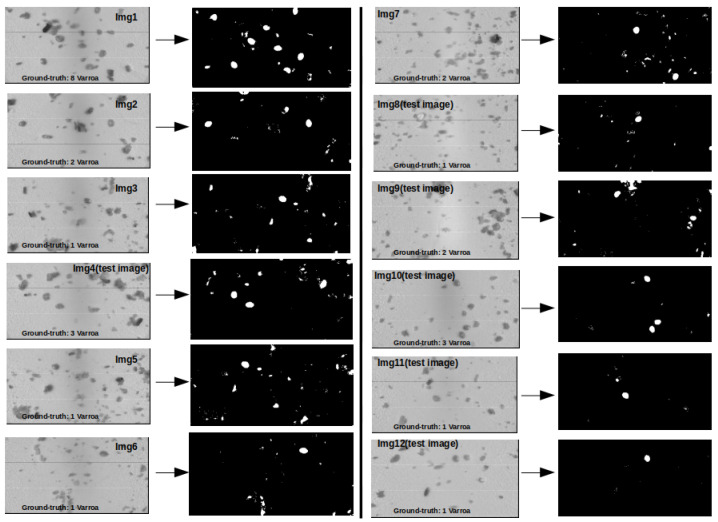
Segmentation output: Varroa mite segmentation achieved using kNN classifier, distinguishing Varroa mites from other class.

**Figure 9 sensors-24-04437-f009:**
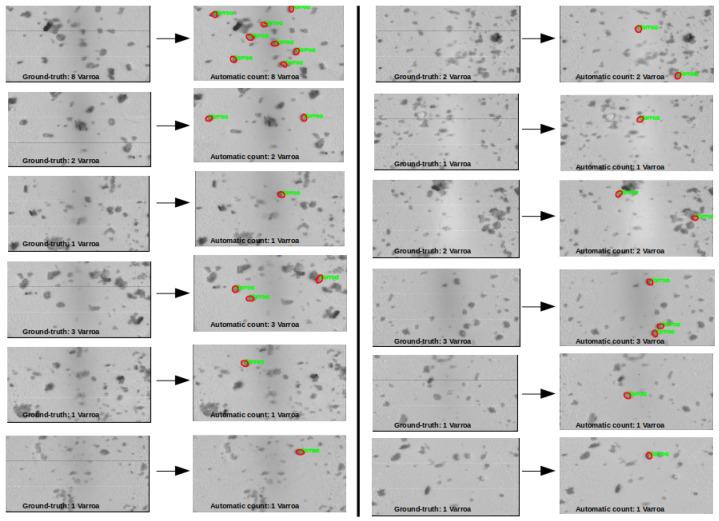
Varroa mite counting results for images 1 to 12: The counts are as follows—8, 2, 1, 3, 1, 1, 2, 1, 2, 3, 1, and 1, respectively. The red circles in the picture indicate the identified Varroa mites. These counts are coinciding with the ground-truth data.

**Figure 10 sensors-24-04437-f010:**
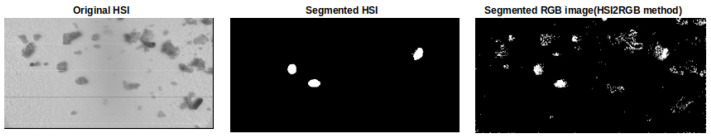
The visual comparisons of the segmentation results between the hyperspectral imaging (HSI) and RGB-like images are presented. The first image represents the original grayscale hyperspectral image, while the middle image illustrates the segmented hyperspectral image. The rightmost image displays the segmented image obtained from the RGB-like image. This comparison highlights the efficacy of utilizing fine-grained spectral composition in achieving well-defined Varroa shapes, contrasting with RGB-like channels that exhibit numerous misclassifications.

**Table 1 sensors-24-04437-t001:** Summary of specifications of the [MV1-D2048x1088-HS02-96-G2-10] hyperspectral sensor.

Image Sensor Specifications	
Manufacturer/type	IMEC, CMV2K-SM5x5
Technology	CMOS
Optical format	2/3″
Optical diagonal	12.76 mm
Resolution	2048 × 1088
Pixel size	5.5 μm × 5.5 μm
Active optical area	11.26 mm × 5.98 mm
Dark current	125e-/s
Read-out noise	13e-
Full well capacity/SNR	11ke- / 105:1
Spectral range	Hyperspectral: 665 to 975 nm (25 pass bands)
Responsivity	Hyperspectral: 454 × 10^3^ DN / (J/m^2^) @ 715 nm/8 bit
Quantum efficiency	Hyperspectral: <18%
Optical fill factor	42% without microlenses
Dynamic range	60 dB
Characteristic curve	Linear, piecewise linear
Shutter mode	Global shutter
**Camera Specifications**	
Interface	GigE
Frame rate	42 fps
Pixel clock	48 MHz
Camera taps	2
Grayscale resolution	8 Bit/10 Bit
Fixed pattern noise (FPN)	<1DN RMS @ 8 Bit
Exposure time range	13 μs–349 ms
Analog gain	yes
Digital gain	0.1 to 15.99 (FineGain)
Trigger modes	Free running (non triggered), external trigger, SWTrigger
Features	Configurable region of interest (ROI), up to 8 regions of interest (MROIs), binning for data preprocessing, decimation in y-direction, 2 look-up tables (12-to-8 Bit) on user-defined image region (Region-LUT), constant frame rate independent of exposure time, crosshairs overlay on the image, temperature monitoring of camera, camera information readable over SDK, ultra low trigger delay and low trigger jitter, extended trigger input and strobe output functionality, status line in picture
Operation temperature/moisture	0 °C … + 50 °C/20% … 80%
Storage temperature/moisture	−25 °C … 60 °C/20% … 95%
Power supply	+12 VDC (−10%) … +24 VDC (+10%)
Power consumption	<5.1 W
Lens mount	C-Mount (CS-Mount optional)
I/O inputs	2× Opto-isolated 2× RS-422 Opto-isolated
I/O outputs	2× Opto-isolated
Dimensions	55 × 55 × 52 mm^3^
Mass	265 g
Connector I/O (power)	Hirose 12-pole (mating plug HR10A-10P-12S)
Connector interface	RJ-45
Conformity	CE / RoHS / WEEE
IP code	IP40

**Table 2 sensors-24-04437-t002:** Machine learning classifier comparison for segmentation. Grid search with cross-validation has been used for hyperparameter selection.

Classifiers	Training Set (Accuracy)	Test Set (Accuracy)	Hyperparameters	F1-Score
“Linear” SVM	0.9932	0.9927	“C”: 1	0.9581
“Rbf” SVM	0.912	0.915	“C”: 0.1, “gamma”: 0.1	0.0
kNN	0.999	0.999	“$n_{neighbors}$”: 1	0.9947
ANN *	0.991	0.992	“$batch_{size}$”: 16, “epochs”: 10	0.995

* Binary Classifier.

**Table 3 sensors-24-04437-t003:** Machine learning classifier comparison for shape analysis. Grid search with cross-validation used for hyperparameter selection.

Classifiers	Test Set (Accuracy)	Cross-Validation (Accuracy)	Hyperparameters
“Linear” SVM	1.0	1.0	“C”: 1
kNN	1.0	0.9784	“$n_{neighbors}$”: 3, “weights”: uniform
Random Forest	1.0	0.9895	“$n_{estimator}$”: 10, “$max_{depth}$”: none
DecisionTree	0.9583	0.9895	“$max_{depth}$”: 5, “$min_{samples_{l}eaf}$”: 2

**Table 4 sensors-24-04437-t004:** Comparative evaluation metrics for segmentation methods based on hyperspectral imaging (HSI) and RGB-like images, providing insights into superior performance of HSI.

Method	Accuracy (%)	Recall (%)	Precision (%)	F1-Score (%)
HSI	99.9	99.17	99.78	99.47
HSI2RGB	98.8	92.93	93.2	93.06

## Data Availability

Data presented in this review is contained within the article. The dataset used in this study is available and could be shared. The dataset includes all necessary files and metadata to replicate the study’s results. The software, including preprocessing scripts and model training codes. Detailed usage instructions are provided to facilitate replication and further research. This transparency ensures that our results can be independently verified and built upon by the research community.

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
