# Peer review of "Varroa Mite Counting Based on Hyperspectral Imaging"

_sensors, 2024, doi:10.3390/s24144437_

Round 1
Reviewer 1 Report
Comments and Suggestions for Authors
The main focus of this article is to investigate the application of Hyperspectral Imaging (HSI) technology and machine learning techniques for the purpose of quantifying varroa mites. There are some recommendations for authors.
1. Are the samples used in the study representative enough of the parasitic situation of dust mites in different environments?
2. What is the correction matrix in section 3, please explain in detail.
3. Is it appropriate to use manual mask labeling for data annotation? Is there any possibility of missed detection?
4. "This dataset divided twice using train-test split function, to extract 10% of the dataset then the second splitting used on this 10% to allocate 80% for training and 20% for testing." Why choose 10% of the data for analysis?
5. Why choose 'Linear' SVM and KNN for segmentation and shape analysis? How can these two machine learning models be used for segmentation?
Reviewer 2 Report
Comments and Suggestions for Authors
This paper describes the use of image processing and machine learning on hyperspectral imaging data to count Varroa mites parasitizing honeybees. This is not a practical study processing large amounts of data in the field, but a preliminary study on a limited amount of data in the laboratory. The research topic is socially and industrially important, and the study of applying hyperspectral sensors to the problem is technically interesting. Unfortunately, however, this paper is not of sufficient quality, as described below.
1. According to the journal's Instructions for Authors, all manuscripts must consist of Introduction, Materials & Methods, Results, Discussion, and Conclusions. This paper does not follow that requirement, and in particular the boundaries between materials and results are not clear. The authors should carefully read the Instructions for Authors and reorganize the manuscript.
2. In the introduction of this paper, the importance and novelty of this study is not clearly stated. While introducing previous studies in detail, the authors should clearly state what makes this study different from those studies and emphasize the significance of those differences.
3. This paper does not adequately describe the experimental equipment. The specific model number of the hyperspectral sensor is given, but that alone does not adequately describe the sensor. It is necessary to explain in detail the principle of operation and performance of the device and, if necessary, summarize the specifications in a table.
4. The paper also does not adequately explain the object being measured. The paper does not explain what a Varroa board is, making it difficult for the reader to understand how it is collected or created.
5. The study applies a variety of image processing and machine learning techniques, but for each technique, there is no description of the algorithm or software used in the paper.
6. The paper has several figures whose contents are difficult to understand, and the captions are poorly explained. For example, Figure 6 has too small labels for the scales and axes. Furthermore, the colors of the items are reversed in Figures 5 and 6, which may mislead the reader. All figures and captions should be revised for greater clarity.
7. The article is inadequate in many respects to describe the research, resulting in a small manuscript page count. The journal recommends that “Article” be more than 16 pages, but this manuscript is only 12 pages, which corresponds to the “Communication” type.
Round 2
Reviewer 2 Report
Comments and Suggestions for Authors
The manuscript was properly revised according to the reviewers' comments and its quality was greatly improved.
On the other hand, one point still unclear in this manuscript is the need to perform hyperspectral imaging in the near-infrared region instead of the visible region for Varroa counting. In general, hyperspectral imaging is performed over a wide range of regions from the visible to the near-infrared. The manuscript does not clearly state the reason for limiting hyperspectral imaging to the near-infrared region. Even if the necessity is obvious from past studies, the background should be explained in the manuscript.
Author Response
The final request of reviewer 2 has been honored by additional information entered in line 55 - 60.
